# Lie Algebra Convolutional Networks with Automatic Symmetry Extraction

## Abstract

Existing methods for incorporating symmetries into neural network architectures require prior knowledge of the symmetry group. We propose to learn the symmetries during the training of the group equivariant architectures. Our model, the Lie algebra convolutional network (L-conv), is based on infinitesimal generators of continuous groups and does not require discretization or integration over the group. We show that L-conv can approximate any group convolutional layer by composition of layers. We demonstrate how CNNs, Graph Convolutional Networks and fully-connected networks can all be expressed as an L-conv with appropriate groups. By allowing the infinitesimal generators to be learnable, L-conv can learn potential symmetries. We also show how the symmetries are related to the statistics of the dataset in linear settings. We find an analytical relationship between the symmetry group and a subgroup of an orthogonal group preserving the covariance of the input. Our experiments show that L-conv with trainable generators performs well on problems with hidden symmetries. Due to parameter sharing, L-conv also uses far fewer parameters than fully-connected layers.

Many machine learning (ML) tasks involve data from unfamiliar domains, which may or may not have hidden symmetries. While much of the work on equivariant neural networks focuses on equivariant architectures, the ability of the architecture to discover symmetries in a given dataset is less studied. Convolutional Neural Networks (CNN) (LeCun et al., 1989; 1998) incorporate translation symmetry into the architecture. Recently, more general ways to construct equivariant architectures have been introduced (Cohen & Welling, 2016a;b; Cohen et al., 2018; Kondor & Trivedi, 2018). Encoding equivariance into an ML architecture can reduce data requirements and improve generalization, while significantly reducing the number of model parameters via parameter sharing (Cohen et al., 2019; Cohen & Welling, 2016b; Ravanbakhsh et al., 2017; Ravanbakhsh, 2020). As a result, many other symmetries such as discrete rotations in 2D (Veeling et al., 2018; Marcos et al., 2017) and 3D (Cohen et al., 2018; Cohen & Welling, 2016a) as well as permutations (Zaheer et al., 2017) have been incorporated into the architecture of neural networks.

Many existing works on equivariant architectures use finite groups such as permutations in Hartford et al. (2018) and Ravanbakhsh et al. (2017) or discrete subgroups of continuous groups, such as 90 degree rotations in (Cohen et al., 2018) or dihedral groups $D_N$ in Weiler & Cesa (2019). Ravanbakhsh (2020) also proved a universal approximation theorem for single hidden layer equivariant neural networks for Abelian and finite groups. General principles for constructing group convolutional layers were introduced in Cohen & Welling (2016b), Kondor & Trivedi (2018), and Cohen et al. (2019), including for continuous groups. A challenge for implementation is having to integrate over the group manifold. This has been remedied either by generalizing Fast Fourier Transforms (Cohen et al., 2018), or using irreducible representations (irreps) (Weiler et al., 2018a) either directly as spherical harmonics as in Worrall et al. (2017) or using more general Clebsch-Gordon coefficients (Kondor et al., 2018). Other approaches include discretizing the group as in Weiler et al. (2018a;b); Cohen & Welling (2016a), or solving constraints for equivariant irreps as in Weiler & Cesa (2019), or approximating the integral by sampling (Finzi et al., 2020).

The limitations in all of the approaches above are that: 1) they rely on knowing the symmetry group *a priori*, and 2) require encoding the whole group into the architecture. For a continuous group, it is not possible to encode all elements and we have to resort to discretization or a truncated sum over irreps. Our work attempts to resolve the issues with continuous groups by using the Lie algebra (the linearization of the group near its identity) instead of the group itself. Unlike the Lie group which is

infinite, the Lie algebra usually has a finite basis (notable exception being Kac-Moody Lie algebras for 2D Conformal Field Theories (Belavin et al., 1984) in physics). Additionally, we show that the Lie algebra basis can be learned during training, or through a separate optimization process. Hence, our architecture, which generalizes a group convolutional layer, is potentially capable of learning symmetries in data without imposing inductive biases.

Learning symmetries in data was tackled in restricted settings of mostly commutative Lie groups as in Cohen & Welling (2014) and 2D rotations and translations in Rao & Ruderman (1999) and Sohl-Dickstein et al. (2010) or permutations (Anselmi et al., 2019). However, the symmetries learned by the architecture are not necessarily familiar spatial symmetries. As we show in the case of linear regression, the symmetries may correspond to transformations preserving the statistics of the data. Specifically, we show a general relation between the symmetries of linear regression and a deformed orthogonal group preserving the covariance matrix. Such symmetries of the probability distribution and ways to incorporate them into the architecture were also discussed in Bloem-Reddy & Teh (2019). The work that is closest in spirit and setup to ours is Zhou et al. (2020) which uses meta-learning to automatically learn symmetries. Although the weight-sharing scheme of Zhou et al. (2020) and their encoding of the symmetry generators is different, their construction does bear some resemblance to ours and we will discuss this after introducing our architecture.

**Contributions** Our main contributions can be summarized as follows

- We propose a group equivariant architecture using the Lie algebra, introducing the Lie algebra convolutional layer (**L-conv**).

- In L-conv the Lie algebra generators can be trained to discover symmetries, and it outperforms CNN on domains with hidden symmetries, such rotated and scrambled images.

- Group convolutional layers on connected Lie groups can be approximated by multi-layer L-conv, and Fully-connected, CNN and graph convolutional networks are special cases of L-conv.

- In linear regression, we show analytical relations between symmetries in and orthogonal groups preserving covariance of data.

## 1 Equivariance in Supervised Learning

Consider the functional mapping $\boldsymbol{y}_i = f(\boldsymbol{x}_i)$ of inputs $\boldsymbol{X} = (\boldsymbol{x}_1, \ldots, \boldsymbol{x}_n)$ to outputs $\boldsymbol{Y} = (\boldsymbol{y}_1, \ldots, \boldsymbol{y}_n)$. We assume each input $\boldsymbol{x} \in \mathbb{R}^{d \times m}$ where $\mathbb{R}^d$ are the "space" dimensions and $\mathbb{R}^m$ the "channels", and $\boldsymbol{y} \in \mathbb{R}^c$ (or $\mathbb{Z}_2^c$ for categorical variables). We assume a group $G$ acts only on the space factor ($\mathbb{R}^d$, shared among channels) of $\boldsymbol{x}$ through a $d$-dimensional representation $T_d : G \to \mathrm{GL}_d(\mathbb{R})$ mapping each $g$ to an invertible $d \times d$ matrix. The map $T_d$ must be continuous and satisfy $T_d(u)T_d(v) = T_d(uv)$ for all $u, v \in G$ (Knapp, 2013, IV.1). Similarly, let $G$ act on $\boldsymbol{y}$ via a $c$-dimensional representation $T_c$. To simplify notation, we will denote the representations simply as $u_\alpha \equiv T_\alpha(u)$. A function $f$ solving $\boldsymbol{y}_i = f(\boldsymbol{x}_i)$ is said to be equivariant under the action of a group $G$ by representations $T_c, T_d$ if

$$u_c \boldsymbol{y} = u_c f(\boldsymbol{x}) = f(u_d \boldsymbol{x}) \qquad \forall u \in G$$
$$\Leftrightarrow f(\boldsymbol{x}) = u_c f(u_d^{-1} \boldsymbol{x}). \tag{1}$$

**Lie Groups and Lie Algebras** The full group of invertible $d \times d$ matrices over $\mathbb{R}$ is the general linear group, denoted as $\mathrm{GL}_d(\mathbb{R})$. It follows that every real $d$- dimensional group representation $T_d(G) \subset \mathrm{GL}_d(\mathbb{R})$. If $T$ is a "faithful representation" (i.e. $T(u) \neq T(v)$ if $u \neq v$), then $G \subset \mathrm{GL}_d(\mathbb{R})$. In our problem, we only know the group $G$ through its linear action on $\boldsymbol{X}$ or on output $h^l$ of layer $l$ in a neural network. Therefore we may assume the representation $T_{d_l}$ acting on $h^l$ with largest $d_l$ is faithful. $\mathrm{GL}_d(\mathbb{R})$ is a Lie group and all of its continuous subgroups $G$ are also Lie groups. Next, we will first briefly review the Lie algebra of a Lie group and then use it to introduce our equivariant architecture.

**Notation** Unless stated or obvious, $a$ in $A^a$ is an index, not an exponent. We write matrix products as $A \cdot B \equiv \sum_a A^a B_a$ Recall that $\boldsymbol{x} \in \mathbb{R}^{d \times m}$. For a linear transformation $A : \mathbb{R}^{d_1} \to \mathbb{R}^{d_2}$

acting on the spatial index or the channel index, we will use one upper and one lower index as in $A_\nu^\mu h_\mu = [A \cdot h]_\nu$. We will use $(a, b, c)$ for channels, and $(\mu, \nu, \rho)$ for spatial, and $(i, j, k)$ for Lie algebra basis indices. We will occasionally keep explicit summation $\sum_i$ for clarity.

For any Lie group $G \subset \mathrm{GL}_d(\mathbb{R})$, group elements infinitesimally close to the identity element can be written as $u = I + \epsilon^i L_i$. We can find a basis $L_i$ which span the "Lie algebra" $\mathfrak{g}$ of the group $G$, meaning they are closed under commutations

$$[L_i, L_j] = L_i L_j - L_j L_i = \sum_k f_{ij}{}^k L_k \tag{2}$$

with $f_{ij}{}^k$ called the structure constants of the Lie algebra. The $L_i$ are called the infinitesimal generators of the Lie group. They define vector fields spanning the tangent space of the manifold of $G$ near its identity element $I$. For elements $u \in G_I$ in the connected component $G_I \subset G$ containing the identity, an exponential map $\exp : \mathfrak{g} \to G_I$ can be defined such that $u = \exp[t \cdot L]$. For matrix groups, if $G$ is connected and compact, the matrix exponential defined through a Taylor expansion is such a map and it is surjective. For most other groups (except $\mathrm{GL}_d(\mathbb{C})$ and nilpotent groups) it is not surjective. Nevertheless, for any connected group ( including $G_I$) every element $u$ can be written as a product $u = \prod_\alpha \exp[t_\alpha \cdot L]$ using the matrix exponential (Hall, 2015). We will use this fact to modify existing results about group equivariant architectures to introduce L-conv.

## 2 GROUP EQUIVARIANT ARCHITECTURE

Consider the case where the solution $f$ to $\boldsymbol{y} = f(\boldsymbol{x})$ is implemented as a feedforward neural network. Denote the linear output of layer $l$ by $h_l = F_l(h_{l-1}) \in \mathbb{R}^{d_l \times m_l}$, with $h_0 = x_i$. Kondor & Trivedi (2018); Cohen & Welling (2016a) showed that a feedforward neural network is equivariant under the action of a group $G$ if and only if each of its layers implement a group convolution (G-conv) given by

$$h_l = (f_l * g_l)(h_{l-1}) = \int_G d\mu(u) f_l(u_{l-1}^{-1} h_{l-1}) g_l(u) \tag{3}$$

where $d\mu(u)$ is the Haar measure on the group manifold, $u_l$ is an appropriate $d_l$-dimensional representation of $G$, and $g_l : G \to \mathbb{R}^{m_l \times m_{l+1}}$ is a set of convolution kernels, where $m_l$ and $m_{l+1}$ are the number of input and output channels (filters). Here $f_l$ are point-wise activation functions. In the formalism of Kondor & Trivedi (2018), we may consider $h_l$ to be a map from the homogeneous space $\mathcal{X}_l$ to $\mathbb{R}^{m_l}$ where $\mathcal{X}_l$ is a space such that the space of maps $L_\mathbb{R}[\mathcal{X}_l]$ is parameterized by $\mathbb{R}^{d_l}$. To use equation 3 in practice, we need to make the integral over the group manifold tractable. Although, Finzi et al. (2020), which approximates the integral over $G$ by sampling, approach the idea of working in the Lie algebra, they use a logarithm map to define the Lie algebra. We take a different approach and formulate the architecture in terms of the Lie algebra basis. We also allow the architecture to learn basis $L_i$ for the symmetries.

For continuous symmetry groups we will now show that using the Lie algebra can approximate G-conv without having to integrate over the full group manifold, thus alleviating the need to discretize or sample the group. The key point is the discreteness and finiteness of the Lie algebra basis for many Lie groups.

**Proposition 1.** *Let $G$ be a Lie group and $T_d : G \to \mathbb{R}^{d_l}$ a $d_l$-dimensional representation of $G$. If a convolution kernel $g_l : G \to \mathbb{R}^{m_{l-1} \times m_l}$ has support only on an infinitesimal $\eta$ neighborhood of identity, a G-conv layer of equation 3 can be written in terms of the Lie algebra.*

*Proof:* Linearization over $\eta$ yields $u_l \approx I + \epsilon \cdot L$ in equation 3, with $L_i \in R^{d_l \times d_l}$. The inverse is then $u_l^{-1} \approx I - \epsilon \cdot L + O(\epsilon^2)$. Since $g_l$ has support only in an $\eta$ neighborhood of identity, fixing a basis $L_i$, we can reparametrize $g_l(I + \epsilon \cdot L) = \tilde{g}_l(\epsilon)$ as a function over the Lie algebra $\tilde{g}_l : \mathfrak{g} \to \mathbb{R}^{m_l \times m_{l+1}}$. Dropping the layer index $l$ for brevity, we have

$$(f * g)(h) \approx \int_G \tilde{g}(\epsilon) f\left((I - \epsilon \cdot L) h\right) d^{n_L} \epsilon \approx \left(W^0 I - W \cdot L\right) h \cdot \nabla_h f(h) \tag{4}$$

$$W^0 \equiv \int_G \tilde{g}(\epsilon) d^{n_L} \epsilon, \qquad W^i \equiv \int_G \epsilon^i \tilde{g}(\epsilon) d^{n_L} \epsilon. \tag{5}$$

where the number of $W^i$ is $n_L$, the number of $L_i$. When $f(h)$ is a homogeneous function like ReLU or linear activation, $h \cdot \nabla_h f(h) = af(h)$. In general, since we are designing the layers, we can choose $h \cdot \nabla_h f(h) = \sigma(h)$ to be a desired activation function. $\qquad\square$

Note that in G-conv, the convolutional kernels $g(u)$ are the trainable weights. The difficulty with learning them is that one has to parametrize and integrate the convolution over the group $G$. The major advantage of working in the Lie algebra instead of $G$ is that in equation 4 the integrals of $g(\epsilon)$ separate from $f(h)$, leaving us with a set of integrated weights $W$. This means that instead of the architecture needing to implement the integral $\int_G g(u)f(uh)d\mu$, we only need to learn a finite number of convolutional weights $W$, like the filter weights in CNN. Additionally, we can learn $L_i$. Indeed, having learnable $L_i$ to learn hidden symmetries is one of main features we are proposing, as we show in our experiments.

**L-conv Layer** The construction in equation 4 is at the core of our results. $h \in \mathbb{R}^{d_{l-1} \times m_{l-1}}$ has components $h_\mu^a$, where $\mu \in \{1, \cdots, d_{l-1}\}$ is the spatial index and $a \in \{1, \cdots m_{l-1}\}$ the channel index. We define the Lie algebra convolutional layer (L-conv) as

$$\text{L-conv}: \quad F_L(h) = \sigma\left(\sum_{i=0}^{n_L} L_i \cdot h \cdot W^i + b\right)$$

$$F_L(h)_\mu^a = \sigma\left(\sum_{\nu,c,i} [L_i]_\mu^\nu h_\nu^c \left[W^i\right]_c^a + b^a\right) \tag{6}$$

where $\sigma$ is the activation function, $L_0 = I$ and $W^i \in \mathbb{R}^{m_l \times m_{l-1}}$ are the convolutional weights, and the second line shows the components (all symbols are indices, not exponents). Note that, similar to discussion in Weiler et al. (2018a), an arbitrary nonlinear activation $\sigma$ may not keep the architecture equivariant under $G$, but linear activation remains equivariant. Also, note that in equation 3, $f_l$ is the activation $\sigma_{l-1}$ of the previous layer. In defining L-conv we are including the activation $\sigma$ after the convolution, which is why $\sigma$ appears outside in equation 6 instead of as $\sigma(h)$.

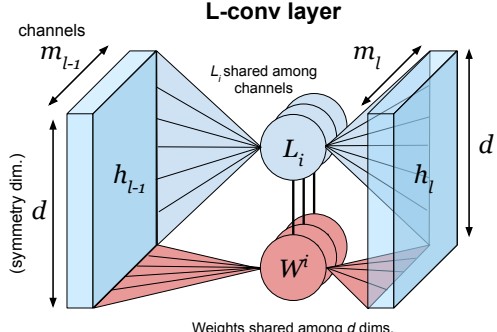

Figure 1: L-conv layer architecture. $L_i$ only act on the $d$ spatial dimensions, and $W^i$ only act on the $m_l$ feature. For each $i$, this action is analogous to a Graph Convolutional Network with $d$ nodes and $m_l$ features per node.

Since in connected Lie groups larger group elements can be constructed as $u = \prod_\alpha \exp[t_\alpha \cdot L]$ (Hall, 2015) from elements near identity, it follows that any G-conv layer can be constructed from multiple L-conv layers on these groups.

**Proposition 2.** *Any G-conv layer on a connected Lie group can be approximated by a multi-layer L-conv to arbitrary accuracy.*

*Proof:* To show this we need to take two steps. First, we discretize the support of $g(u)$ in equation 3 to infinitesimally small sets $G = \bigcup_k G_k$, each having support around an element $u_k \in G_k$. This allows us to write $(f * g)(h) \approx \sum_k \Delta\mu_k g(u_k) f(u_k^{-1}h)$, where $\Delta\mu_k = \int_{G_k} d\mu$. Next, we write each $u_k$ as a product. On connected groups for any $u_k \in G$, we can write $u_k = \prod_{\alpha=1}^l v_\alpha$ where $v_\alpha = \exp[t_\alpha \cdot L] \approx (I + t_\alpha \cdot L)$ are elements close to identity. Then, $f(uh)$ can be written as a composition $l$ G-conv layers, each with weights $g_\alpha(u) = \delta(u - v_\alpha)$ (defined using a faithful representation of $G$)

$$f(u_k^{-1}h) = f\left(\prod_\alpha v_\alpha h\right) = f(F_l \circ \cdots \circ F_1(h)), \qquad F_\alpha(h) = \int_G d\mu(u) f_\alpha(u_\alpha h) g_\alpha(u) \tag{7}$$

Since $u_\alpha$ are near identity, each $F_\alpha$ can be converted to an L-conv layer with linear activation and weights $W = t_\alpha$ and $W^0 = 1$. Thus, the G-conv layer can be approximated as sum and composition

of L-conv layers. Each $G_k$ can be considered one output channel for the multi-layer L-conv and $g(u_k)$ are weights of a final aggregation layer.

$\square$

So far we have shown that L-conv layers could replace G-conv for connected groups. The advantage of using L-conv layers is that for many potential symmetry groups, such as orthogonal and unitary groups, the number of generators $L_i$ is finite and often a small number of them may be sufficient. For instance, consider $SO(2)$ rotations on $p \times p$ images. The flattened image has $d = p^2$ dimensions and the number of $\mathrm{GL}_d(\mathbb{R})$ generators is $p^4$. However the induced representation of $SO(2)$ on $\mathbb{R}^d$ has a single generator. In fact, in many domains we expect symmetries hidden in the data to be much lower in dimensionality than the data itself. Additionally, in the other extreme, where there is no restriction on the symmetry, meaning $G = \mathrm{GL}_d(\mathbb{R})$, we observe that L-conv becomes a fully-connected layer, as shown next.

**Proposition 3.** *A fully-connected neural network layer can be written as an L-conv layer using* $\mathrm{GL}_d(\mathbb{R})$ *generators, followed by a sum pooling and nonlinear activation.*

*Proof:* The generators of $\mathrm{GL}_d(\mathbb{R})$ are one-hot $\boldsymbol{E} \in \mathbb{R}^{d \times d}$ matrices $L_i = \boldsymbol{E}_{(\alpha,\beta)}$ which are non-zero only at index $i = (\alpha, \beta)$ [1] with elements written using Kronecker deltas

$$\mathrm{GL}_d(\mathbb{R}) \text{ generators} : L_{i,\mu}^\nu = [\boldsymbol{E}_{(\alpha,\beta)}]_\mu^\nu = \delta_{\mu\alpha}\delta_\beta^\nu \tag{8}$$

Now, consider the weight matrix $w \in \mathbb{R}^{m \times d}$ and bias $b \in \mathbb{R}^m$ of a fully connected layer acting on $h \in \mathbb{R}^d$ as $F(h) = \sigma(w \cdot h + b)$. The matrix element can be written as

$$w_b^\nu = \sum_\mu \sum_{\alpha,\beta} w_b^\alpha \mathbf{1}_\beta [\boldsymbol{E}_{(\alpha,\beta)}]_\mu^\nu = \sum_\mu \sum_{\alpha,\beta} W_{(\alpha,\beta)}^{b,1} [\boldsymbol{E}_{(\alpha,\beta)}]_\mu^\nu = \sum_\mu W^{b,1} \cdot [L]_\mu^\nu \tag{9}$$

meaning an L-conv layer with weights $W_{(\alpha,\beta)}^{b,1} = w_b^\alpha \mathbf{1}_\beta$ (1 input channel, and $\mathbf{1}$ being a vector of ones) followed by pooling over $\mu$ is the same as a fully connected layer with weights $w$. $\square$

Thus, interestingly, more restricted groups, rather than large, meaning groups with fewer $L_i$, lead to more parameter sharing. Indeed, as we show below, the most restricted L-conv are graph convolutional networks. It is also easy to see that familiar CNN can be encoded as L-conv. Note that, generally shift operators used in CNN are thought of as group elements for a discrete permutation group. However, the same shift operators $L_i$ can be used to create continuous fractional shifts as $(I + \alpha L_i)/(\alpha + 1)$. Since shifts commute with each other, they form a basis for the Lie algebra of an Abelian subgroup of $\mathrm{GL}_d(\mathbb{R})$.

**Proposition 4.** *CNN is a special case of L-conv where the generators are shift operators.*

*Proof:* In 1D CNN, for a kernel size $k$ we have $k$ shift operators $L_i$ given by $L_{i\mu}^\nu = \delta_{\mu,\nu-i}$. Plugging this into equation 6 and doing the sum over $\nu$, we recover the convolution formula

$$F(h)_\mu^a = \sigma\left(\sum_{i,c} h_{\mu-i}^c W_c^{i,a} + b^a\right) \tag{10}$$

In higher dimensional CNN, $i$ covers the relevant flattened indices for the required shifts. For instance, a $p \times q$ image, when flattened becomes a vector of length $pq$. To cover a $(2, 2)$ convolutional kernel, we need shifts to be by $0, 1, q, q+1$ pixels. Thus, we need four $L_i$ given by

$$L_{i\mu}^\nu = \delta_{\mu,\nu-s_i}, \qquad\qquad s_i = [0, 1, q, q+1]_i \tag{11}$$

This means that the number of generators $L_i$ is the size of the convolutional kernel in CNN. Generalization to higher dimensions is straightforward. $\square$

In CNN, covering larger patches is equivalent to $g(u)$ covering a larger part of the group. But large convolutional kernels are not generally used. Instead, we achieve larger receptive fields by adding more CNN layers, similar to Proposition 2.

---

[1] We may also label them by single index like $i = \alpha + \beta d$, but two indices is more convenient.

**Connection with Graph Convolutional Networks** Finally, we note that the L-conv layer equation 6 has the structure of a Graph Convolutional Networks (GCN) Kipf & Welling (2016) with multiple propagation rules $L_i$ derived from graph adjacency matrices and $\mu, \nu$ indexing graph vertices. Thus, an L-conv with a single $L$ is a GCN. In fact, there is a deeper connection here, as we show now.

**Proposition 5.** *A GCN with propagation rule $A \in \mathbb{R}^{d \times d}$ is equivalent to an L-conv, with the group being 1D flows generated by $A - I$.*

*Proof:* We can define a linear flow similar to a diffusion equation

$$h(t + \delta t) = A h_\nu(t) \qquad\qquad \Rightarrow \frac{dh(t)}{dt} = \lambda(A - I)h(t) \qquad (12)$$

where $\lambda = \delta t / dt$ sets the time scale. Thus $L = \lambda(A - I)$ is the generator a 1D group of flows with elements $u = \exp[tL]$, a subgroup of the group of diffeomorphisms on $\mathbb{R}^d$, with a single generator $L$. Thus, a GCN with propagation rule $A$ is an L-conv using $L_0 = I$ and $L_1 = L$ with the same convolutional weights for $L_0, L_1$. $\qquad\square$

Hence, the most restricted L-conv based on 1D flow groups with a single generator are GCN. These flow groups include Hamiltonian flows and other linear dynamical systems. CNN can also be interpreted as a GCN with multiple propagation rules, where each shift operator is a subgraph of the grid network.

## 3 LEARNING POTENTIAL SYMMETRIES

When dealing with unknown continuous symmetry groups, it can be virtually impossible to design a G-conv, as it relies on the structure of the group. The Lie algebra, however, has a much simpler, linear structure and universal for all Lie groups. Because of this, L-conv affords us with a powerful tool to probe systems with unknown symmetries. L-conv is a generic weight-sharing ansatz and the number of $L_i$ can be expected to be small in many systems. This means that even if we do not know $L_i$, it may be possible to *learn* the $L_i$ from the data. In fact, as we show in our experiments, learning low-rank $L_i$ via SGD simultaneously with other weights yield impressive performance on data with hidden symmetries (Fig. 2), without needing any input about the symmetry.

**Learning the $L_i$** We learn the $L_i$ using SGD, simultaneously with $W^i$ and all other weights. Our current implementation is similar to a GCN $F(h) = \sigma(L_i \cdot h \cdot W^i + b)$ where both the weights $W^i$ and the propagation rule $L_i$ are learnable. When the spatial dimensions $d$ of the input $x \in \mathbb{R}^{d \times c}$ is large, e.g. a flattened image, the $L_i$ with $d^2$ parameters can become expensive to learn. However, generators of groups are generally very sparse matrices. Therefore, we represent $L_i$ using low-rank decomposition $L_i = U_i V_i$. An encoder $V$ of shape $n_L \times d_h \times d$ encodes $n_L$ matrices $V_i$, and a decoder $U$ of shape $n_L \times d \times d_h$. Here $d$ is the input dimensions and $d_h$ the latent dimension with $d_h \ll d$ for sparsity. In order for the $L_i$ to form a basis for a Lie algebra, they should be closed under the commutation relations equation 2, as well as orthogonal under the Killing Form (Hall, 2015, Chapter 6). These conditions can be added to the model as regularizers (see Appendix F.2), but regularization also introduces an additional time complexity of $O(n_L^2 d_h^2 d)$, which can be quite expensive compared to the $O(n_l d_h d)$ of learning $L_i$ via SGD. Therefore, in the experiments reported here we did not use any regularizers for $L_i$.

**Comparison with Meta-learning Symmetries by Reparameterization (MSR)** Recently Zhou et al. (2020) also introduced an architecture which can learn equivariances from data. We would like to highlight the differences between their approach and ours, specifically Proposition 1 in Zhou et al. (2020). Assuming a discrete group $G = \{g_1, \ldots, g_n\}$, they decompose the weights $W \in \mathbb{R}^{s \times s}$ of a fully-connected layer, acting on $\boldsymbol{x} \in \mathbb{R}^s$ as $\text{vec}(W) = U^G v$ where $U^G \in \mathbb{R}^{s \times s}$ are the "symmetry matrices" and $v \in \mathbb{R}^s$ are the "filter weights". Then they use meta-learning to learn $U^G$ and during the main training keep $U^G$ fixed and only learn $v$. We may compare MSR to our approach by setting $d = s$. First, note that although the dimensionality of $U \in \mathbb{R}^{nd \times d}$ seems similar to our $L \in \mathbb{R}^{n \times d \times d}$, the $L_i$ are $n$ matrices of shape $d \times d$, whereas $U$ has shape $(nd) \times d$ with many more parameters than $L$. Also, the weights of L-conv $W \in \mathbb{R}^{n \times m_l \times m_{l-1}}$, with $m_l$ being the number of channels, are generally much fewer than MSR filters $v \in \mathbb{R}^d$. Finally, the way in which $Uv$ acts on data is different from L-conv, as the dimensions reveal. The prohibitively high dimensionality of $U$ requires MSR to adopt a sparse-coding scheme, mainly Kronecker decomposition. Though not necessary, we too

choose to use a sparse format for $L_i$, finding that very low-rank $L_i$ often perform best. A Kronecker decomposition may bias the structure of $U^G$ as it introduces a block structure into it.

**Contrast with Augerino** In a concurrent work, (Benton et al., 2020) propose Augerino, a method to learn invariances with neural networks. Augerino uses data augmentation to transform the input data, which means it is restricting the group to be a subgroup of the augmentation transformations. The data augmentation is written as $g_\epsilon = \exp\left(\sum_i \epsilon_i \theta_i L_i\right)$ (equation (9) in Benton et al. (2020)), with randomly sampled $\epsilon_i \in [-1, 1]$. $\theta_i$ are trainable weights which determine which $L_i$ helped with the learning task. However, in Augerino, $L_i$ are fixed *a priori* to be the six generators of affine transformations in 2D (translations, rotations, scaling and shearing). In contrast, our approach is more general. We learn the generators $L_i$ directly without restricting them to be a known set of generators. Additionally, we do not use the exponential map, hence, implementing L-conv is very straightforward. Lastly, Augerino uses sampling to effectively cover the space of group transformations. Since the sum over Lie algebra generators is tractable, we do not need to use sampling.

Next, we show that in certain cases, such as linear regression, the symmetry group can be derived analytically. While the results may not apply to more non-linear cases, they give us a good idea of the nature of the symmetries we should expect L-conv to learn.

## 3.1 Symmetries of Linear Regression

In the case of linear regression we can derive part of the symmetries explicitly, as shown next. We absorb biases into the regression weights $\boldsymbol{A}$ as the last row and append a row of $1$ to the input.

**Theorem 1.** *Consider a linear regression problem on inputs $\boldsymbol{X} \in \mathbb{R}^{d \times n}$ and labels $\boldsymbol{Y} \in \mathbb{R}^{c \times n}$ like above. We are looking for the linear function $\boldsymbol{Y} = \boldsymbol{A}\boldsymbol{X}$. For this problem is equivariant under a group $G$, through two representations $u \in T_d(G)$ and $u_c \in T_c(G)$ acting on $\boldsymbol{X}$ and $\boldsymbol{Y}$, respectively, it is sufficient for $u$ and $u_c$ to satisfy*

$$u\boldsymbol{H}u^T = \boldsymbol{H} \qquad\qquad u_c\boldsymbol{Y}\boldsymbol{X}^T u^T = \boldsymbol{Y}\boldsymbol{X}^T. \qquad (13)$$

*where $\boldsymbol{H} \equiv \frac{1}{n}\boldsymbol{X}\boldsymbol{X}^T$ is the covariance matrix of the input.*

*Proof:* The equivariance condition equation 1 becomes

$$u_c\boldsymbol{Y} = u_c\boldsymbol{A}\boldsymbol{X} = \boldsymbol{A}u\boldsymbol{X} \qquad\qquad \Rightarrow \boldsymbol{A} = u_c\boldsymbol{A}u^{-1} \qquad (14)$$

Assuming that the number of samples is much greater than features, $n \gg d$, and unbiased data, $\boldsymbol{X}^T\boldsymbol{X}$ will be full rank $d \times d$ and that its inverse exists. The solution to the linear regression $\boldsymbol{Y} = \boldsymbol{A}\boldsymbol{X}$ is given by $\boldsymbol{A} = \boldsymbol{Y}\boldsymbol{X}^T(\boldsymbol{X}\boldsymbol{X}^T)^{-1}$. Using the definition of covariance $\boldsymbol{H}$ above, the condition equation 14 becomes $\boldsymbol{A} = u_c\boldsymbol{Y}\boldsymbol{X}^T u^T(u\boldsymbol{X}\boldsymbol{X}^T u^T)^{-1}$. Thus, a sufficient condition for equation 14 to hold is equation 13. $\qquad\square$

The first condition in equation 13 is unsupervised, stating that $u$ preserves the covariance $\boldsymbol{H}$, while the second condition is supervised, stating that $u_c$ and $u$ preserve the cross-correlation of input and labels $\boldsymbol{Y}\boldsymbol{X}^T$. Next, we show that the group satisfying equation 13 is related to an orthogonal group.

**Corollary 1.** *The subgroup of symmetries of linear regression satisfying $u\boldsymbol{H}u^T = \boldsymbol{H}$ in equation 13 form an orthogonal group isomorphic to $SO(d)$, with a Lie algebra basis given by*

$$L_i = \boldsymbol{H}^{1/2}L_i'\boldsymbol{H}^{-1/2} \qquad\qquad L_i' \in so(d) \qquad (15)$$

*Proof:* multiplying $u\boldsymbol{H}u^T = \boldsymbol{H}$ by $\boldsymbol{H}^{-1/2}$ from both sides we see that $u' \equiv \boldsymbol{H}^{-1/2}u\boldsymbol{H}^{1/2}$ satisfies $u'u'^T = I$ meaning $u' \in SO(d)$. Expanding elements near identity we get $u \approx I + \epsilon \cdot L = \boldsymbol{H}^{1/2}(I + \epsilon' \cdot L')\boldsymbol{H}^{-1/2}$. Setting $\epsilon = \epsilon'$ proves equation 15. (see Supp D.2 for more details) $\qquad\square$

## 4 Experiments

To understand precisely how L-conv performs in comparison with CNN and other baselines, we conduct a set of carefully designed experiments. Defining pooling for L-conv merits more research. Without pooling, we cannot use L-conv in state-of-the-art models for familiar problems such as image

classification. Therefore, we use the simplest possible models in our experiments: one or two L-conv, or CNN, or FC layers, followed by a classification layer.

**Test Datasets** We use four datasets: MNIST, CIFAR10, CIFAR100, and FashionMNIST. To test the efficiency of L-conv in dealing with hidden or unfamiliar symmetries, we conducted our tests on two modified versions of each dataset: 1) **Rotated:** each image rotated by a random angle (no augmentation); 2) **Rotated and Scrambled:** random rotations are followed by a fixed random permutation (same for all images) of pixels. We used a 80-20 training test split on 60,000 MNIST and FashionMNIST, and on 50,000 CIFAR10 and CIFAR100 images. Scrambling destroys the correlations existing between values of neighboring pixels, removing the locality of features in images. As a result, CNN need to encode more patterns, as each image patch has a different correlation pattern.

**Test Model Architectures** We conduct controlled experiments, with one (Fig. 2) or two (Fig. 3) hidden layers being either L-conv or a baseline, followed by a classification layer. For CNN, L-conv and L-conv with random $L_i$, we used $n_f = m_l = 32$ for number of output filters (i.e. output dimension of $W^i$). For CNN we used $3 \times 3$ kernels and equivalently used $n_L = 9$ for the number of $L_i$ in L-conv and random L-conv. As described in sec. 3, we encode $L_i$ as sparse matrices $L_i = U_i V_i$ with hidden dimension $d_h = 16$ in Fig. 2 and $d_h = 8$ in Fig. 3, showing that very sparse $L_i$ can perform well. The weights $W^i$ are each $m_l \times m_{l+1}$ dimensional. The output of the L-conv layer is $d \times m_{l+1}$. As mentioned above, we use two FC baselines. The FC in Fig. 2 and FC($\sim$L-conv) in Fig. 3 mimic L-conv, but lacks weight-sharing. The FC weights are $W = ZV$ with $V$ being $(n_L d_h) \times d$ and $Z$ being $(m_{l+1} \times d) \times d_h$. For "FC (shallow)" in Fig. 3, we have one wide hidden layer with $u = n_{L-conv}/(mdc)$, where $n_{L-conv}$ is the total number of parameters in the L-conv model, $m$ and $c$ the input and output channels, and $d$ is the input dimension. We experimented with encoding $L_i$ as multi-layer perceptrons, but found that a single hidden layer with linear activation works best. We also conduct tests with two layers of L-conv, CNN and FC (Fig. 3), with each L-conv, CNN and FC layer as descried above, except that we reduced the hidden dimension in $L_i$ to $d_h = 8$.

**Baselines** We compare L-conv against three baselines: CNN, random $L_i$, and fully connected (FC). Using CNN on scrambled images amounts to using poor inductive bias in designing the architecture. Similarly, random, untrained $L_i$ is like using bad inductive biases. Testing on random $L_i$ serves to verify that L-conv's performance is not due to the structure of the architecture, and that the $L_i$ in L-conv really learn patterns in the data. Finally, to verify that the higher parameter count in L-conv is not responsible for the high performance, we construct two kinds of FC models. The first type ("Fully Conn." in Fig. 2 and "FC ($\sim$ L-conv)" in Fig. 3) is a multilayer FC network with the same input ($d \times m_0$), hidden ($k \times n_L$ for low-rank $L_i$) and output ($d \times m_1$) dimensions as L-conv, but lacking the weight-sharing, leading to much larger number parameters than L-conv. The second type ("FC (shallow)" in Fig. 3) consists of a single hidden layer with a width such that the total number of model parameters match L-conv.

**Results** Fig. 2 shows the results for single layer experiments. On all four datasets both in the rotated and the rotated and scrambled case L-conv performed considerably better than CNN and the baselines. Compared to CNN, L-conv naturally requires extra parameters to encode $L_i$, but low-rank encoding with rank $d_h \ll d$ only requires $O(d_h d)$ parameters, which can be negligible compared to FC layers. We observe that FC layers consistently perform worse than L-conv, despite having much more parameters than L-conv. We also find that not training the $L_i$ ("Rand L-conv") leads to significant performance drop. We ran tests on the unmodified images as well (Supp. Fig 4), where CNN performed best, but L-conv trails closely behind CNN.

Additional experiments testing the effect of number of layers, number of parameters and pooling are shown in Fig. 3. On CIFAR100, we find that both FC configurations, FC($\sim$L-conv) and FC(shallow) consistently perform worse than L-conv, evidence that L-conv's performance is *not* due to its extra parameters. L-conv outperforms all other tested models on rotated and scrambled CIFAR100. Without pooling, we observe that both L-conv and CNN do not benefit from adding a secnd layer. This can be explained by Proposition 2, which states that multi-layer L-conv is still encoding the same symmetry group $G$, only covering a larger portion of $G$. Our hypothesis is that lack of pooling is the reason behind this, as we discuss next.

**Pooling on L-conv** On Default CIFAR100, CNN with Maxpooling significantly outperforms regular CNN and L-conv, indicating the importance of proper pooling. Interestingly, on rotated and scrambled

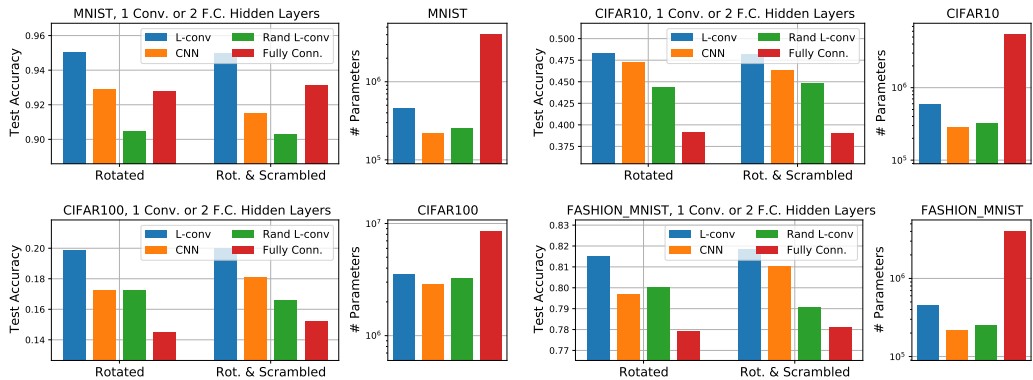

Figure 2: Results on four datasets with two variant: "Rotated" and "Rotated and scrambled". In all cases L-conv performs best. On MNIST, FC and CNN come close, but using 5x more parameters.

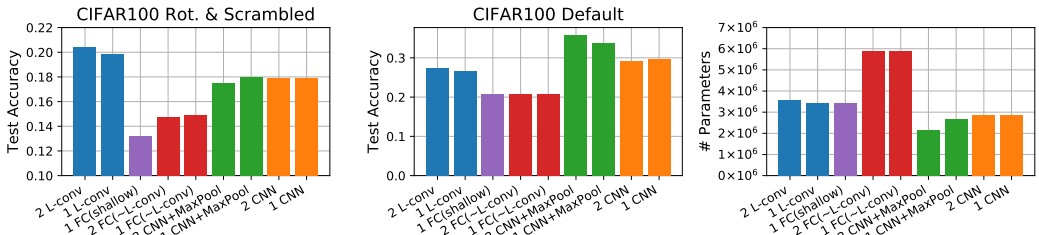

Figure 3: Comparison of one and two layer performance of L-conv (blue), CNN without pooling (orange), CNN with Maxpooling after each layer (green), fully connected (FC) with structure similar to L-conv (red) and shallow FC, which has a single hidden layer with width such that the total number of parameters matches L-conv (purple). The labels indicate number of layers and layer architecture (e.g. "2 L-conv" means two layers of L-conv followed by one classification layer). Left and middle plots show test accuracies on CIFAR100 with rotated and scrambled images, and on the original CIFAR100 dataset, respectively. The plot on the right show the number of parameters of each model, which is the same for the two datasets.

CIFAR100, we find that max-pooling does not yield any improvement. We believe the role of pooling is much more fundamental than simple dimensionality reduction. On images, pooling with strides blurs our low-level features, allowing the next layer to encode symmetries at a larger scale. Cohen & Welling (2016a) showed a relation between pooling and coset of subgroups and that strides are subsampling the group to a subgroup $H \subset G$, resulting in outputs which are equivariant only under $H$ and not the full $G$. These subgroups appearing at different scales in the data may be quite different. However, a naive implementation of pooling on L-conv may involve three $L_i$ and be quite expensive (see Appendix C). Devising an efficient and mathematically sound pooling algorithm for L-conv is a future step we are working on.

## 5 DISCUSSIONS

We introduced the L-conv layer, a group equivariant layer based on Lie algebras of continuous groups. We showed that many familiar architectures such as CNN, GCN and FC layers can be understood as L-conv. L-conv is easy to setup and can learn symmetries during training. On domains with hidden symmetries we find that an L-conv layer outperforms other comparable baseline layer architectures. L-conv can in principle be inserted anywhere in an architecture. L-conv can also be composed in multiple layers, though Proposition 2 suggest it would be approximating the same symmetry group, and we did not observe significant improvements in performance in two layer tests. However, we believe this is due to lack of pooling for L-conv. For CNN on images, coarse-graining (maxpooling with strides) allow the system to find features at a different scale, whose symmetry may be a subset of the lower scale (Cohen & Welling, 2016a) or perhaps new symmetries. Thus, one future work

direction is defining proper pooling (see Supp C). Lastly, to ensure $L_i$ behave like a Lie algebra basis, we need to include regularizers enforcing orthogonality among the $L_i$, which is another future direction.

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
