# OpenReview forum: "Lie Algebra Convolutional Neural Networks with Automatic Symmetry Extraction"
_ICLR.cc/2021/Conference — Reject_

### Official Review · AnonReviewer4 · 2020-10-26
**An interesting approach**

**Rating:** 6
**Confidence:** 5

**Review:**

### Summary
The paper presents a new approach for building group-equivariant neural networks. The authors propose *L-conv*, a layer which is equivariant to transformations from a group $G$ in the neighborhood of identity. They show that a network with a sufficient number of such layers is $G$-equivariant as a whole. Additionally, the authors propose to learn the structure of the group directly from the training data instead of incorporating it in the network *a priori*.

### Strengths:
1. The paper is interesting and is easy to follow. The problem the authors are solving is clearly stated from the very beginning and is described in the title.
2. Proposition 1 is insightful. Equations 4 and 5 contain a novel parametrization of convolutional kernels. It also allows considering standard convolution from a different perspective.
3. Proposition 2 reveals a useful connection between the  Lie algebra-Lie group correspondence and a network of *L-conv* layers.

### Weaknesses:
1. **Experiments**.  The conducted experiments do not demonstrate the advantage of the proposed models over other models that use the power of data symmetry. The advantage of the proposed models over conventional CNN is not properly demonstrated.
    * The decision of using very shallow networks is not clear. An experiment with a 1-layer CNN is not directly generalizable to deeper networks.
    * Paragraph **Learning $L_i$ during Training** does not contain enough information for a clear understanding of the chosen models. A diagram of the chosen models or a table could make it more clear.
    * Figure 2, MNIST. The *L-conv* model contains 2 times more parameters than the standard CNN. It demonstrates $\approx 95$ \% while the standard CNN demonstrates $\approx 93$ \%. It is not clear whether the improvement is caused by the increased number of trainable parameters or by the proposed layer. The same argument is applicable to the CIFAR10/100 and the FashionMNIST experiments.
    * There are no demonstrations of the learned generators $L$. A demonstration of it will make it possible to understand whether the learned structure matches the known structure of the data. For example the rotation operator for Rotation MNIST.
    * It is worth understanding whether a network of *L-conv* layers converges to a group equivariant network or proposition 2 is purely theoretical.
2. Paragraph **Notation** and the proposed notation itself do not seem useful.
    * The Einstein summation rule is used only in Equation 4 which can be easily rewritten with $\sum$. The rewritten version will be more readable.
    * Equation 2 is not used in the paper at all.
    * Some letters are used multiple times while meaning different things. For example $f$ is a structure constraint in Equation 2, it is a neural network in paragraph 1 of Section 2, and it is a point-wise activation function later in the same Section.

### Decision
To sum up, the paper proposes an interesting method for building group equivariant neural networks which allows for estimating symmetries from the data instead of incorporating it in the network *a priori*. However, the experimental results are not sufficient for proving the advantage of the proposed approach. The theoretical contribution of the paper is not sufficient for considering it as pure-theoretical. The paper seems raw for acceptance at the current stage.


# Revision
After a fruitful conversation with the authors I am changing my decision.
The conducted experiment presented in Figure 4 confirms an advantage of L-conv.
The paper has a room for improvement. However, it contains valuable discoveries and can be clearly placed in the field of symmetry-aware models. The proposed approach is well-described and is properly evaluated on shallow networks. While the generalization of the results to deeper models is not possible (and the reasons are described by the authors correcly), the approach is interesting by itself and is important to the field.

---

> ### Public Comment · ~Joey_Bose1 · 2020-11-16
> **Review misses key theoretical contributions**
>
> Let me preface this comment by saying I have no knowledge of the authors or any connection with the paper and thus have no stake in its acceptance outside of a general interest in this subfield. I find the review to be a bit on the harsh side and while I agree with the reviewer that the experiments are limited and could be improved the significance of the theoretical result is tremendous. It is not trivial to show that L-conv layers can easily be built and the past prior work Finzi et. al 2020 has significant implementation drawbacks which this paper addresses. Furthermore, the insight that GCN and vanilla CNN layers can be modeled with L-conv layers is a powerful insight that connects useful theory from Lie algebras to practical use cases in geometric deep learning ---i.e. on Graphs.  Indeed, these insights could potentially pave the way for much more powerful GNN's or better inductive biases for arbitrary continuous symmetry groups. Given the potential significance of these results, I would kindly urge the reviewer to reassess the theoretical contributions in this work as in my humble opinion they are significant to the field at large.

---

> > ### Author Response · Authors · 2020-11-24
> > **L-conv unifies and connects mutliple approaches to equivariance**
> >
> > Thank you for your comment.
> > We also believe that the research in equivariance needs to converge to simpler more practical methods which are applicable across a wide variety of problems.
> > L-conv may be demonstrating a fundamental way for how equivariance should be encoded.
> > It is, of course, only a first step. But the simple structure of L-conv and its relation with the zoo of architectures may allow us to understand the better how symmetries should be encoded in the architecture. This can be very useful as an ansatz for modeling physical systems where potential symmetries may exist but we may not have discovered them yet.

---

> > ### Comment · AnonReviewer4 · 2020-11-24
> > **Reply**
> >
> > I have changed my decision about the paper. Thank you for your valuable comment, which helped me to assess the paper more accurately.

---

> ### Author Response · Authors · 2020-11-24
> **L-conv is a new paradigm for equivariance. It needs further development; New experiments added to show L-conv's performance is not due to the number of parameters**
>
> Thank you for your comments.
> We cannot claim that all technical and conceptual issues with L-conv have been resolved with this paper and that it should be able beat all other G-conv approaches.
> Like the humble beginnings of G-conv such as Cohen and Welling "Group Equivariant Convolutional Networks" implementing only a discrete 4-fold rotations, we believe our work can open a new direction for research on equivariance.
> We are doing more experiments to address some of your concerns.
> In what follows we try to clarify some of our choices.
>
>
> ___They show that a network with a sufficient number of such layers is -equivariant as a whole.___
> To clarify, multiple L-conv layers using the same Lie algebra cover a larger part of the group manifold, approximating a G-conv. Each L-conv layer is equivariant because is implements group convolution.
> It just uses a density $g_l(u)$ concentrated near identity.
>
> ___Weaknesses: Experiments ... do not demonstrate the advantage ... over other models that use the power of data symmetry___
> We claim that L-conv could potentially alleviate the need to hard-code the inductive bias when we do not know the symmetry.
>
> However, if in a problem we have extensive theoretical knowledge of potential symmetries, we should use them.
> They may also be encoded as pre-trained $L_i$ and they may outperform $L_i$ learned during SGD.
> L-conv is useful in situations where we do not know all of the symmetries.
>
> ___The advantage [over] CNN is not properly demonstrated___
> CNN implement translational invariance.
> But if the dataset is a set of signals about behaviors of particles, planets, etc. there is no grid over which we could use CNN.
> For example, a flattened, scrambled image with random noise insertions, is not treatable with CNN, neither with any other existing G-conv.
> L-conv, on the other hand, does not care about the structure of the data and can be applied to arbitrary spaces (i.e. arbitrary signals in $\mathbb{R}^{d\times c}$), including graphs.
> Also, while CNN are equivariant under translations, there is no true translational invariance in MNIST (the corners of MNIST images are always empty), only partial reuse of small patterns in different parts of the images.
> Our intuition to use CNN on MNIST does not stem from MNIST, but from our deeper knowledge about images and that patterns may repeat and get shifted.
> It may be that if we train L-conv on a much larger dataset of images and then re-use the learned $L_i$ on MNIST, it performs better.
> What L-conv does is it tries to find the best $L_i$ based on the data, which helps with the ML task at hand.
>
> ___decision of using very shallow networks is not clear.
> An experiment with a 1-layer CNN is not directly generalizable to deeper networks___
> We used shallow networks for two reasons.
> First, we wanted a bare-bones model to contrast the difference between using a CNN layer vs L-conv.
> Therefore, we didn't add FC layers after the convolutions.
> Second reason for not using multi-layer L-conv is that we lack a maxpooling layer (see edits to the text), which is a future direction.
> It is easy to build multi-layer L-conv and we have added those experiments (Fig 3).
> But without pooling, a two-layer L-conv or CNN only yielded small improvement.
> We believe Proposition 2 provides a possible explanation.
> Multi-layer L-convs without pooling should be learning the same symmetry group.
> We have added a discussion on this to the paper.
>
> ___Learning___ $L_i$ ___during Training [not] enough information [on] models.
> A diagram [or] table could make it more clear___
> Thank you, we have expanded that part and added details of the parameters and configuration.
>
> ___L-conv ... 2 times more parameters than CNN [95% vs 93%] not clear whether ... caused by the increased number of trainable parameters___
>
> L-conv naturally has more parameters than CNN because in CNN $L_i$ are preset and fixed, whereas they are learned in L-conv.
> We can also pretrain $L_i$.
> FC models (red bars in Fig. 2) have much more parameters than L-conv, but do perform much worse.
> Also, if $L_i$ are fixed, L-conv has the same number of parameters as CNN.
> Random L-conv (Fig. 2 green ) have $L_i$ fixed to random matrices and perform much worse than L-conv, showing the value of learning $L_i$.
>
> ___no demonstrations of the learned___ $L$ ___... whether [it] matches the known structure of the data ... rotation operator___
> We have added visualizations of $L_i$ for MNIST in Appendix B. The eigenvectors of $LL^T$ mimic the structure of data, but they are different from PCA (eigenvectors of the covariance matrix).
>
> ___It is worth understanding whether a network of L-conv layers converges to a group equivariant network or proposition 2 is purely theoretical___
> We think they should and that proposition 2 is not purely theoretical.
> Testing this with fixed $L_i$ for a specific group is rather straightforward.
> We are working on this for future steps.

---

> > ### Comment · AnonReviewer4 · 2020-11-24
> > **Answer**
> >
> > Thank you for your reply!
> >
> > ### Number of parameters
> > Figure 2.
> > It is worth understanding whether the proposed model is more accurate because of the increased number of parameters or because of the proposed structure. Currently, it is not demonstrated. Comparing to a CNN with 2x 3x fewer parameters does not make any sense. A simple question is still not answered: **"Is your model better just because it has more parameters?"**
> > The experiment with L_conv rand demonstrates the only thing "built-in random symmetry is worse than translation symmetry as in CNN". This is _de facto_ true
> >
> > ### Learning the symmetries
> > L_conv is developed to learn the symmetries of the dataset. Translation is not the only symmetry of CIFAR or MNIST.
> > Thus, if L_conv can learn the symmetries it should learn them.
> > If we take a look at Figure 4 we see that L_conv is worse than the CNN when no transformation is applied to the dataset.
> > After it is transformed, both accuracies drop significantly. A reasonable conclusion from the presented results is that L_conv does not learn translation and/or extra transformations. It is simply better in some settings just because the model is 2x 3x times bigger than the CNN.
> >
> > At the current stage, the proposed model **could** and **should** learn the symmetries as it is described by the authors. But the sufficient evidence of this is not demonstrated.
> >
> > ### Text updates
> > The authors adjusted the text according to my notes. Additionally, they have fixed some moments which improved the readability of the text.
> >
> > ### To sum up
> > The authors propose a linear layer of a special structure. The current experiments do not demonstrate that it indeed learns the symmetries of the training dataset. The theoretical part does not prove the advantage of the proposed layer, as well.
> > I urge the authors to conduct more solid experiments and to prove that the proposed method works as it is supposed to work.
> > Currently, the claims of the paper are ambitious but the experimental results **do not confirm them**.
> > I think that currently the paper is below the high standards of ICLR.

---

> > > ### Author Response · Authors · 2020-11-24
> > > **Figure 3 are the new experiments, demonstrating it is *not* the number of parameters.**
> > >
> > >
> > > ### Parameters: Will the Reviewer please comment on FC results being much worse than L-conv despite having more parameters?
> > > Can you please also comment on Figure 3?
> > > There we are showing two types of FC models, both of which despite having the same or more parameters than L-conv perform much worse.
> > > __FC(shallow)__ matching precisely the number of parameters in L-conv
> > > __FC(~L-conv)__ which has the same encoder as L_i, but the decoder is matching the output of L-conv layer, leading to much larger number of parameters.
> > >
> > > As we stated, you cannot use the fact L-conv with the same number of filters and kernel size as CNN has more parameters as an argument to dismiss it. L-conv is learning the L_i whereas CNN has fixed L_i. And the FC experiments demonstrate it is not the number of parameters.
> > >
> > > If the reviewer does not find the FC experiments satisfactory, can they please let us know what experiment would convince them it is not the parameter count?
> > > As for Figure 2, we have a choice for how sparsely we encode L-conv. In Fig 3, we chose $d_h=8$ for the encoding dimension of $L_i$ to reduce the parameter count compared to Fig 2, which used $d_h=16$. Still, you see that L-conv performs better than CNN.
> > >
> > >
> > > ### Symmetries
> > > There is no translational symmetry in the MNIST, not the other image datasets: no shifted version of the same image exists in MNIST. This is a misconception about CNN, because we always think of the receptive field of a neuron, rather than the output of a filter over a whole image. This would merit a much longer discussion, not suitable to be done here.
> > > The symmetries which may be hidden in them can be complex and can only be inferred from the distribution of the data (see Theorem 1). We think that L-conv is doing the best job it could with the information present in the dataset.
> > >
> > > And again, it is not due to the parameters, because FC has even more parameters but does not yield the same performance.

---

> > > > ### Comment · AnonReviewer4 · 2020-11-24
> > > > **An experiment**
> > > >
> > > > Thank you for your reply.
> > > >
> > > > A comparison between a CNN and an L-conv net with the same number of parameters will be a good demonstration. In the literature, usually it is achieved by varying the number of channels in each layer.

---

> > > > > ### Author Response · Authors · 2020-11-24
> > > > > **L-conv still performs better than CNN with matching parameters on Rot. scrambled MNIST**
> > > > >
> > > > > Thank you for the great suggestion.
> > > > > Please find Figure 4 in the updated supplement. We find that L-conv still performs better.
> > > > > You may also run the notebook `L-conv-extra-exp-2020-11-24.ipynb`
> > > > > in the code folder to experiment with this.
> > > > >
> > > > > Is there any other experiments you would like to see?
> > > > > Thank you.

---

> > > > > > ### Comment · AnonReviewer4 · 2020-11-24
> > > > > > **Good result**
> > > > > >
> > > > > > Thank you for your quick response!
> > > > > > The result is good.
> > > > > > I need to reassess the paper, because for me this experiment indeed demonstrates the advantage of the proposed methods over standard CNNs.

---

> ### Author Response · Authors · 2020-11-25
> **Thank you**
>
> Thank you very much for giving us a better chance and affording us with the opportunity to share this work with the community. We hope it can foster new discussions and debates on equivariance. As you correctly point out, many hurdles remain. We are passionate about this project mainly because we feel there is deep, fundamental science in equivariance in ML and believe we need a simpler, unified language to understand it.

---

### Official Review · AnonReviewer3 · 2020-10-28
**Learning symmetries through Lie-algebra convolutions**

**Rating:** 8
**Confidence:** 4

**Review:**

Summary:

The paper meticulously builds a theoretical framework for Lie-algebra convolutional layers and then goes on to show how CNNs, GCNs and FC layers are a special case of L-convs. The paper also demonstrates how the underlying generators can be learnt from data and provide convincing supporting experimental results. The proposed L-conv layers also use much fewer parameters compared to previous works.

Key strengths:

The theoretical framework developed in this paper, starting from Lie groups and equivariance and invariance definitions is very elegant and convincing. I checked the maths at each step and am convinced that it is correct, to the best of my knowledge. I did need to refer to Hall 2015 though. Intuitively as well as mathematically, it makes sense to me. The comparison to MSR (Zhou 2020) seems fair to me.
The experiments, though limited, in the main paper, are quite convincing. The experiments are cleverly constructed and provide enough justification to support the utility of L-conv layers in comparison to CNN and FC layers.

Questions:

For Figure 2: For CIFAR100 and FashionMNIST, CNN seems to do better on "rotated+scrambled" compared to "rotated". What is the reason behind that? This is not seen in any other method or dataset.

Suggestions for improvements:

1. The paper inherently assumes familiarity with Lie groups/Lie algebra or even exp/log of matrices which I am familiar with, but not all readers will be. Therefore, instead of citing Hall 2015, it would be good to cite Sections within the textbook. This will aid uptake of an important mathematical sub-field.
2. There is no mention of accompanying code in the manuscript. Would the authors consider making it available upon acceptance? It would help further research in this area.
3. The section on linear regression (3.1) seems to occur again in an expanded form in the supplementary material (Sec C). The derivation in the supplementary material was a little bit clearer.
4. It would be worthwhile checking the paper for typos. A few that I noted: larest, wight, "a too many"
5. Figure 1 is not easy to interpret. A substantial caption would be beneficial. It's also not referenced in the text.

Overall comments:
The main contribution of this paper is the development of the theoretical framework for Lie-algebra convolutions. The paper does so very convincingly and I regard this as an important contribution to the area of deep learning. This may open the door to the field using the correct inductive biases for many problems in vision, speech and physics. I enjoyed reading this paper, including the supplementary material that makes a start on imposing orthogonality during regularization for complex-valued neural networks.

---

> ### Author Response · Authors · 2020-11-24
> **Code is now shared in supplement; Figure 1 updated; discussion of Lie algebra regularizers added**
>
> Thank you, we are glad you liked our paper.
> About your comments:
>
> ___For Figure 2: For CIFAR100 and FashionMNIST, CNN seems to do better on "rotated+scrambled" compared to "rotated"___
> We are running more tests to see if this is a robust effect or within the margin of error.  Maybe the rotations are serving as partial augmentation and the scrambling is dispersing the low-level features over a wider area of the image for fashion_mnist, which unlike MNIST has a lot of small features.
>
> ___citing Hall 2015 ... cite Sections___ Added.
>
> ___code... making it available?___
> yes of course. We are sorry that we forgot to include it in the supplement.
> We have done so now.
>
> ___linear regression (3.1) [expanded] derivation in supplementary... clearer.___
> Yes.
> We felt that part didn't fit well with the flow of the paper.
> So to save space, we moved it later.
> The goal of section 3.1 was to derive conditions for what the $L_i$ should be.
> We are trying to see if 3.1 can be made clearer.
>
> ___Figure 1 is not easy to interpret. Caption, reference in text___
> Thank you for clarifying.
> We have modified it and added caption to it.
>
> ___imposing orthogonality during regularization for complex-valued neural networks.___
> Yes, in fact, we also worked on a separate optimization process solely to find a Lie algebra basis, imposing orthogonality and allowing complex values.
> We were able to learn generators for $SO(3), SU(2), SO(3,1)$ and more, starting from the metric they preserve.
> We did not use such regularizers in the experiments reported in the paper, mainly due to their computational complexity.
> We have added a discussion of this in the __Learning__ $L_i$ paragraph.

---

### Official Review · AnonReviewer2 · 2020-10-29
**An interesting description of group equivariance with Lie algebras.**

**Rating:** 7
**Confidence:** 4

**Review:**

This paper describes an approach to making deep learning robust to arbitrary symmetries. To this end, the authors introduce a group equivariance architecture of neural networks which ensures that the function learned by the network has the equivariance property with respect to the symmetry group action (roughly speaking, it ensures that the symmetry affects the input and output of the function in the same way, or that the application of the symmetry does not influence the how the function works).

The building block of the group equivalence architecture is the L-conv layer, i.e. a layer which models a Lie algebra. The L-conv layer works somewhat like the convolutional layer, but instead of using the kernel mechanism, it multiplies the previous layer by L_i which is the basis spanning the Lie algebra around the corresponding Lie group’s identity element. The authors relate to the condition (provided in existing literature) for a feedforward network layer to be equivariant under a group action of G, and show that the L-conv layer satisfies this condition (Propositions 1 and 2).

Subsequently, it is shown that fully connected layers, convolutional layers as well as graph convolutional networks can be expressed with L-conv layers. Further, the authors compare their approach to a very similar one submitted concurrently to ICLR. They also consider the special case of linear regression and show that the symmetry Lie group can be described analytically. Finally, experiments comparing the L-conv based architecture to CNNs and fully connected layers on standard deep learning benchmark data subject to rotation and random permutations are presented.


*****Strengths:*****

Modelling arbitrary symmetries and equivariance / invariance relations with continuous groups is an important topic that has sparked much interest recently in various application areas from computer vision to computational chemistry. The work presented in this paper is certainly relevant.

I found the paper well organised and easy to follow.

While the approach presented here is similar to a couple of papers reviewed in related work (especially Zhou et al.), I think that the description of the symmetry relations in the language of Lie groups and Lie algebra bases is novel and well presented here.


*****Weaknesses:*****

The main weakness of this paper, in my view, is that no description of L_i learning is provided. To some extent, I am not sure if I understand the process correctly after reading the paper. While the theory appears to guarantee that correctly trained L_i are able to produce L-conv layers acting as fully connected layers, convolutional layers or graph convolution networks while ensuring the equivariance property, no analysis is given as to how feasible it is to find the correct basis L_i. In my understanding, the entire contribution of the paper hinges on the ability to find L_i and thus it should be given more space in the paper. The only comment on this is given in the experiment section, where the authors mention an additional autoencoder being employed. Meanwhile, from section 2 and equation 6, one could guess that L_i could be learned directly by sgd. If the latter is not the case, I think a discussion as to why not would make the paper clearer. If there are other theoretical results on the structure / properties of Lie algebra bases that the authors make use of, I think they should be provided too.

In the experiment section, the only experiment concerns rotation as the modelled symmetry. As the goal of the paper is modelling arbitrary symmetries, i.e. all functions with the equivariance property wrt some Lie group, would it not be more convincing to devise an arbitrary set of transformations (satisfying equation 1) without a simple analytical form (as in the case of rotation) and show that one can guarantee robustness against this arbitrary set? The current experiment could be beaten by any approach parametrising rotational symmetry.


*****Questions / feedback:*****

See weaknesses section.


*****Typos:*****

Theorem 1 (the third sentence) should be rephrased -> for this problem to be equivariant?

Equation 15: so -> SO

Section 5: fing -> find


*****Post Rebuttal*****

I would like to thank the authors for their comments. They clarified most of my doubts regarding the paper, most importantly about learning the basis $L_i$. While I agree with reviewer 3 that there is room for improvement when it comes to experimental evaluation, I appreciate the changes made during the rebuttal period and keep my original score.

---

> ### Author Response · Authors · 2020-11-24
> **$L_i$ are learned through SGD**
>
> Thank you. About your comments:
>
> ___Description of learning___ $L_i$
> We have added more details.
> In short, $L_i$ is learned via SGD, simultaneously with all other weights.
> It is the same as a GCN $\sigma(L_i\cdot h \cdot W^i)$ where  instead of only learning $W^i$, both $W^i$ and $L_i$ are learnable parameters.
>
> ___"no analysis is given as to how feasible it is to find the correct basis___ $L_i$ ___" and "theoretical results on the structure / properties of Lie algebra bases"___
> To ensure $L_i$ would make a proper Lie algebra basis, we may add regularizers to enforce orthogonality and commutation rules among $L_i$.
> For a discussion of these regularizers, see Appendix E and F.
> However, most theoretical results on Lie algebras involve operations between two or three $L_i$ (e.g. commutators, Killing form, Jacobi identity).
> The computational cost of a naive implementation of these regularizers outweighed their benefits in our tests, so we did not use them in our final experiments.
> In terms of feasibility, we find that learning sparse $L_i$ is quite easy and efficient (see edits to "Learning $L_i$ during Training").
>
> Regarding whether $L_i$ are the correct basis:
> Theorem 1 provides a clue as to what the correct symmetries look like, but they are not feasible to use in practice.
> The number of generators of $so(d)$ is quite large at $d(d-1)/2$, but the data may have very few symmetries (i.e. one rotation in Rot. MNIST).
> Therefore we concluded that finding a fixed number of $L_i$ via SGD without regularization was the most efficient way in practice and much simpler to implement.
> We have added a discussion on this in the paragraph on Learning $L_i$ and added a visualization of $L_i$ found for MNIST as well as the top eigenvectors of $\sum_iL_iL_i^T$ in Appendix B.
>
> ___Learning___ $L_i$: ___additional auto-encoder, or learned directly via SGD? Discuss why not___
> They are learned directly via SGD.
> The generators of Lie algebras are very sparse and likely have a low-rank approximation.
> Therefore, we encode $L_i$ through a low-rank tensor decomposition, which is the autoencoder you are referring to.
> We are sorry if the term autoencoder was misleading.
> To be clear, there is no autoencoder in the architecture, only a the low-rank encoding of $L_i$.
> We have rewritten that part of the text to clarify this.
> The low-rank $L_i$ are learned via SGD like regular weights during training.
>
> ___more convincing to devise an arbitrary set of transformations (satisfying equation 1) without a simple analytical form (as in the case of rotation) and show that one can guarantee robustness against this arbitrary set?___
> This is very good question.
> We have not thought about the robustness aspect here.
> Our goal with scrambling images was exactly to show that the symmetry generators can be very unfamiliar and arbitrary.
> Our implicit assumption when training L-conv is that the actual number of symmetries is low (we pick $n_L\ll d^2$).
> However, we did not investigate the robustness theoretically.
> We think that it could be great future direction to investigate whether vanilla SGD will work for arbitrary complex symmetries, or if additional regularization is needed.
> Other experiments could include
> many-body physics problems to see if L-conv could discover symmetries of the low energy Hamiltonian.
> But if our work gets published, perhaps the community would already have problems with hidden symmetries that could benefit L-conv.

---

### Decision · Program_Chairs · 2021-01-07
**Final Decision**

**Decision:**

Reject

**Comment:**

The decision for this paper is quite difficult: the methodological ideas are interesting, such as learning the generators directly, but the experimental results are relatively weak. Perhaps learning the generators is too unconstrained. Moreover, the proposed 'L-conv' builds heavily on prior work such as the 'LieConv' model of Finzi et. al, which is not made very transparent in the current narrative. And LieConv does provide better performance than L-Conv --- this should also be made clear in the text. This was a very difficult call, and after extensive discussion, the decision is intended to be in the long term best interests of the paper. The ideas are interesting and warmly appreciated, the reviewers appreciated aspects of the response, and the project is sufficiently promising that it was felt that their impact would be much greater if the experimental execution were strengthened, such that the project largely terminating at this point would do it a disservice in the long run. There was a general feeling that this is still a 'work in a progress' and was somewhat rush-written. The authors are strongly encouraged to continue pursuing this work, strengthening the experiments and narrative,  as above.

---

> ### Comment · ~Nima_Dehmamy1 · 2021-01-26
> **Final comments and Comparison to LieConv**
>
> We appreciate the AC's comments and the effort of the AC and reviewers in giving us valuable feedback. We will continue the endeavor.
> We would like to add a final comment for the general public.
> 1. Our paper received an average score of 7, which puts it as one of the top 12-25% of accepted papers, and top-4% of all reviewed papers (https://docs.google.com/spreadsheets/d/1n58O0lgGI5kI0QQY9f4BDDpNB4oFjb5D51yMr9fHAK4/edit?usp=sharing)
> 2. We respectfully disagree with the AC's comment regarding LieConv (Finzi et al. 2020).
> As mentioned in our paper,
> A) LieConv uses a predefined Lie algebra basis, whereas we learn the basis.
> B) Our method does not rely on sampling or discretization, whereas LieConv uses Monte Carlo sampling.
> C) Our model is easy to implement, LieConv uses matrix logarithm.
> As the independent comment by Joey Bose to Reviewer 4 stated: "Finzi et. al 2020 has significant implementation drawbacks which this paper addresses." We will add more thorough comparisons with LieConv in a future draft.
>
> We hope that the community finds our work useful and builds upon it.